# MITIGATE POSITION BIAS IN LARGE LANGUAGE MODELS VIA SCALING A SINGLE DIMENSION

## ABSTRACT

Large Language Models (LLMs) are increasingly applied in various real-world scenarios due to their excellent generalization capabilities and robust generative abilities. However, they exhibit position bias, also known as "lost in the middle", a phenomenon that is especially pronounced in long-context scenarios, which indicates the placement of the key information in different positions of a prompt can significantly affect accuracy. This paper first explores the micro-level manifestations of position bias, concluding that attention weights are a micro-level expression of position bias. It further identifies that, in addition to position embeddings, causal attention mask also contributes to position bias by creating position-specific hidden states. Based on these insights, we propose a method to mitigate position bias by scaling this positional hidden states. Experiments on the NaturalQuestions Multi-document QA, KV retrieval, LongBench and timeline reorder tasks, using various models including RoPE models, context window-extended models, and Alibi models, demonstrate the effectiveness and generalizability of our approach. Our method can improve performance by up to 15.2% by modifying just one dimension of hidden states.

## 1 INTRODUCTION

Long-context large language models (LLMs) (Gradient, 2024; Reid et al., 2024; Liu et al., 2024a; Young et al., 2024; Abdin et al., 2024; DeepSeek-AI, 2024) have recently garnered significant attention within the community, enabling LLMs to handle longer and more complex tasks such as long-context question-answering (Caciularu et al., 2023; Li et al., 2024) and repository-level code understanding (Bairi et al., 2023). However, recent researches (Li et al., 2024; Liu et al., 2024b; Li et al., 2023; Shi et al., 2023; Tang et al., 2023), indicates that these long-context LLMs struggle to effectively and consistently utilize all the information provided in the context, exhibiting a position bias known as "lost in the middle", which means LLMs tend to ignore information in the middle of the prompt, even though they can utilize the information at the beginning and end of the prompts well. This issue occurs in nearly all LLMs (Liu et al., 2024b; Junqing et al., 2023; Zhang et al., 2024), whether they are decoder-only models or encoder-decoder models, powerful models or small LLMs. For example, for the GPT-3.5-Turbo model in the NaturalQuestion multi-document QA task, the performance difference between ground-truth information placed in the middle of the prompt versus at the ends is 22 points with 2.3k tokens prompt (Liu et al., 2024b). This significantly impacts the practical application of LLMs in real-world scenarios. Studies (Kamradt, 2023; Zhao et al., 2024) show that this position bias becomes more severe as the context length increases, hindering the practical application of long-context LLMs.

Previous works have analyzed this issue from the perspectives of data distribution (Junqing et al., 2023; Yu, 2023; An et al., 2024) and position embeddings (Zhang et al., 2024; Chen et al., 2023b). For example, FILM (An et al., 2024) addresses position bias by constructing data with key information distributed in various positions for supervised fine-tuning (SFT). Ms-PoE (Zhang et al., 2024) mitigates position bias by interpolating RoPE (Su et al., 2024) using head-wise scaling factors. However, these methods require additional overhead for training or online estimation of scaling coefficients and are currently applicable to only a few models, limiting their generalizability.

To fundamentally understand and alleviate position bias in LLMs, we first explored the micro-level manifestation of position bias in LLMs and observed patterns in the attention weights consistent with

position bias. Next, we investigated the underlying causes of attention weight-induced position bias. By respectively modifying position embedding and causal mask, we found that, in addition to position embedding, the causal mask also significantly affects position bias. Further analysis revealed that the causal mask introduces "positional hidden states", which are positively correlated with absolute positions, thereby conveying positional information to LLMs. These positional hidden states appear regardless of what position encoding method is used, including RoPE (Su et al., 2024), Alibi (Press et al., 2022), and even NoPE (Haviv et al., 2022).

Based on the above findings, we propose a position bias mitigation method named **"scale positional hidden states"**. Specifically, we first design a prior-based searching algorithm that quickly identifies which dimensions of hidden states within the model are positional hidden states, using monotonicity, smoothness, and loss on validation sets as indicators. Next, we design an attention modification algorithm that only let the scaled hidden states influence the attention of the last token of the prompt, efficiently implemented using FlashAttention (Dao, 2023).

Extensive experiments on various models, including LLaMA-2 (Touvron et al., 2023), Vicuna (Chiang et al., 2023), Mistral (Jiang et al., 2023a), Gemma (Team et al., 2024), Qwen (Bai et al., 2023a), and MPT (Team, 2023), and across different tasks, including Multi-document QA, KV retrieval, LongBench (Bai et al., 2023b) benchmark, and the timeline reorder task (Li et al., 2023), demonstrate that our method effectively mitigates position bias by modifying only one dimension of the hidden states of the model, achieving improvements of up to 15.2%. Our method is compatible with various position embeddings, including RoPE (Su et al., 2024) and Alibi (Press et al., 2022), and shows good generalization.

Our main contributions are as follows:

1. We find that position bias can be reflected in attention patterns.

2. We discover that the causal mask also introduces position bias and generates positional hidden states correlated to absolute positions in the hidden layers.

3. We propose a method for identifying and scaling the positional hidden states to mitigate position bias.

## 2 BEYOND POSITION EMBEDDINGS: POSITIONAL INFORMATION CAN BE SEEN IN HIDDEN STATES

In this section, we first identifies patterns in attention weights that closely correspond to position bias. Then, we discover that, apart from position embeddings, position information in the LLMs can also be generated by the causal mask, which tends to accumulate in a few specific hidden states channels and bears significant responsibility for the emergence of position bias.

### 2.1 MICROSCOPIC MANIFESTATIONS OF POSITION BIAS IN TRANSFORMERS: ATTENTION WEIGHT PATTERNS

The attention of auto-regressive can be represented by the following equations:

$$\boldsymbol{q} = \mathcal{P}(W^Q \boldsymbol{h}(n), n), \quad \boldsymbol{k} = \mathcal{P}(W^K \boldsymbol{h}(m), m)$$
$$\boldsymbol{a}_{n,m} = \text{Softmax}(\frac{\boldsymbol{q}\boldsymbol{k}^{\mathrm{T}} + \text{Mask}}{\sqrt{d}}) \tag{1}$$

where $\boldsymbol{h}$ is the hidden states, and $\boldsymbol{h}(n)$ is the hidden state of the n-th token. $W^Q, W^K$ are the weights of the linear layers, $\mathcal{P}$ is the position encoding function like RoPE (Su et al., 2024), $d$ is the dimensionality of query and key states, and $n$ and $m$ are the positional order information. Mask is the causal mask.

To explore the micro-level manifestations of position bias in Transformers, we analyzed the attention weights for sentences containing key information, using a KV retrieval task, which requires the model to retrieval the ground-truth value of the given key from a list containing 50 Key-Value pairs (see Appendix B for details). As shown in Figures 1, in deep layers the model exhibits retrieval-like behavior, focusing on ground-truth information, forming a diagonal pattern observed in Figure 1b.

While in other shallow layers, it always focus most attention on the start or end of the prompt, wherever the key information is located, exhibiting vertical lines patterns, as shown in Figure 1a.

In these layers exhibiting retrieval-like behavior, it can be observed that the attention weights for key information (Gold KV) exhibit patterns similar to position bias: when key information is located at the start or end of the prompt, the attention weights focused on it are relatively higher, while in the middle, they are significantly lower. Moreover, we extract the attention to key information (average of layers 15~25) with different context length in Figure 1c, where as the context length grows, the attenuation of attention weights with respect to position becomes more pronounced, reaching almost zero at the middle. More details about this are in Appendix E and B.

Furthermore, in Appendix 4.3, we found artificially adjusting the attention weights to the key information can directly improve the corresponding accuracy. Thus, we claim that position bias is to a large extent caused by the attention weights patterns at the micro level.

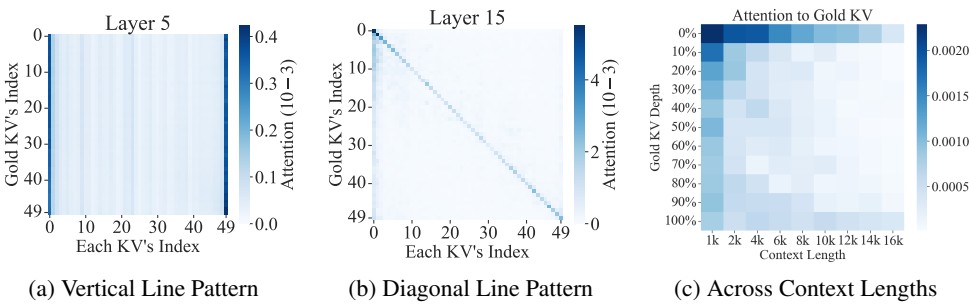

| (a) Vertical Line Pattern | (b) Diagonal Line Pattern | (c) Across Context Lengths |

Figure 1: Attention distribution of the ground-truth KV pair to each KV pair across different positions on the KV retrieval task (Liu et al., 2024b) using Mistral-7B (Jiang et al., 2023a). (a) and (b) show the results averaged across all heads of the layer. (c) shows the attention of the ground-truth KV to the ground-truth KV (i.e., diagonal lines from (b)) across different context lengths.

## 2.2 CAUSAL MASK ALSO CONTRIBUTES TO POSITION BIAS

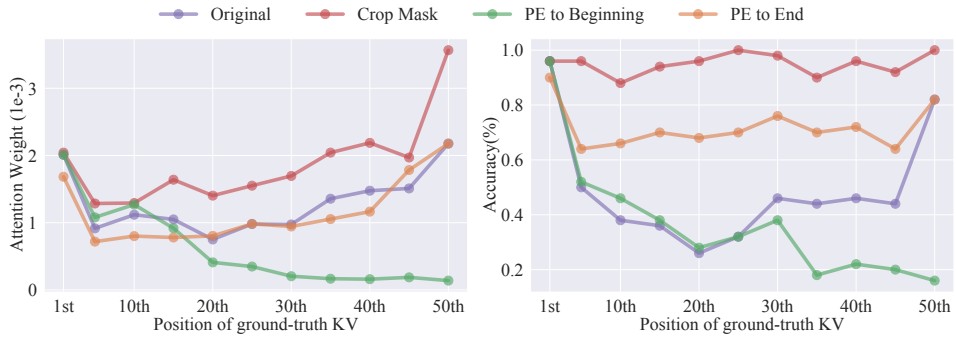

Figure 2: Performance of different methods with the ground-truth KV at different positions in the KV retrieval task (Liu et al., 2024b) using Mistral-7B (Jiang et al., 2023a).

Based on Equ.(1), position embedding $\mathcal{P}$ allows LLMs to acquire postional information. However, recent works (Haviv et al., 2022; Wang et al., 2024; Chi et al., 2023) indicate that, besides position embeddings, the causal mask can also introduce positional information.

In this section, we aim to determine whether these two factors affect position bias by modifying different properties of the ground-truth KV pair. We introduce three baselines: (1) **Crop Mask**, which alters the causal mask so the ground-truth KV pair sees only itself, not previous tokens; (2) **PE to Beginning**, which assigns the position IDs of the ground-truth KV pair to match the first KV pair; (3) **PE to End**, which assigns the position IDs to match the last KV pair. Further details are provided in Appendix C.

As shown in Figure 2, the original results exhibit a "lost in the middle" pattern not only in accuracy but also in attention weight. Secondly, PE to end has a certain degree of help, but can hardly allow the model's performance to match the accuracy when the ground-truth KV pair is positioned at the start or end of the prompt. Furthermore, PE to Beginning results in a noticeable performance drop as well as attention weight reduction when the gold KV is close to the end. In contrast, modifying the casual mask effectively enhances attention, especially to the latter KVs, and let the performance at the middle be improved to almost on par with the beginning. Based on the above observations, we can conclude that besides position embedding, the casual mask is also an important factor affecting position bias as well as corresponding attention weights. Moreover, solely modifying the position embedding hardly alleviates position bias completely.

### 2.3 Casual Mask Stores Position Information in Specific Hidden states Channels

**Definition 2.1** (Positional Hidden States). *Let $h_k(p)$ denote the $k$-th dimension of the hidden states across each token's position $p$. We define positional hidden states $h_t$ as hidden states whose values vary consistently and monotonically with the position sequence. Therefore, their derivative (after curve fitting) should always be positive or negative:*

- $h_t'(p) > 0$, $\forall p$ or $h_t'(p) < 0$, $\forall p$

Previous works (Haviv et al., 2022; Wang et al., 2024; Chi et al., 2023) have found that the positional information generated by causal mask is implicitly stored in hidden states. However, in fact, we find it can be observed explicitly, from "positional hidden states".

To further analyze how positional information is transmitted in transformers, we define a special type of hidden state that directly reflects absolute positional information with high correlations to position IDs, called positional hidden states, as defined in Definition 2.1. We employ monotonicity rather than correlation as the primary property of positional hidden states, as correlation does not account for the sequential nature of positions. As shown in Figure 3, our experiments reveal that causal LLMs consistently possess such hidden states across most layers (details in Appendix F), even though these models do not have explicit absolute position embeddings, which means the causal mask is a very possible factor that provides absolute positional information. To demonstrate that these position hidden states are formed under the influence of the causal mask rather than the position embeddings, we conduct perturbation experiments on the causal mask and position embedding, as shown in Appendix C.

Based on the findings from Section 2.2, we conclude that the causal mask encodes positional information in certain hidden states, which subsequently influences attention weights and introduces position bias.

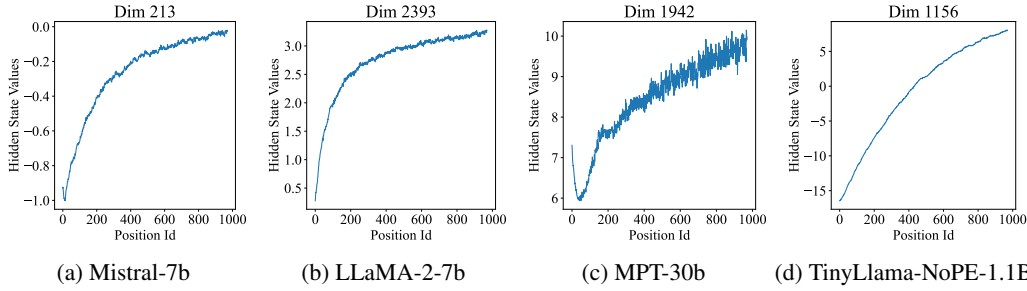

(a) Mistral-7b  (b) LLaMA-2-7b  (c) MPT-30b  (d) TinyLlama-NoPE-1.1B

Figure 3: Averaged positional hidden states across all layers in different models.

## 3 Methodology

Based on the findings in Section 2, although the causal mask profoundly influences position bias, it is not feasible to know the positions of effective information in the prompt in advance, making methods that modify the causal mask difficult to design. Therefore, we propose a method to mitigate position

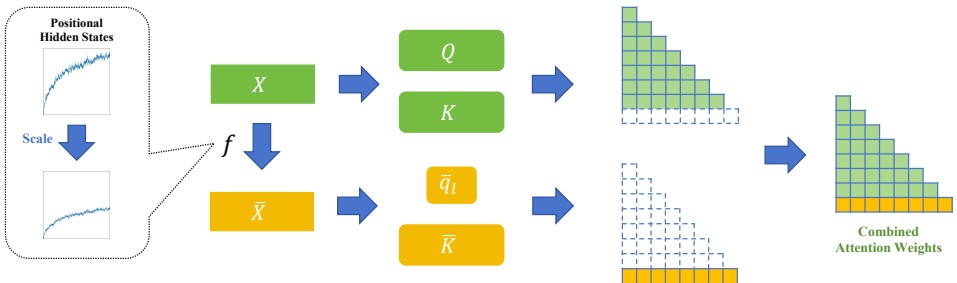

Figure 4: The framework of scaling positional hidden states and modifying attention.

bias by scaling the positional hidden states, as shown in Figure 4. Specifically, it consists of two steps: identifying the positional hidden states $h_t$ and scaling them by the factor $s$.

## 3.1 PROBLEM FORMULATION

Given a pre-trained LLM $\boldsymbol{\theta}$ and a general dataset $\{\boldsymbol{x}, \boldsymbol{y}\}$, our objective is to find the optimal positional hidden states $h_t$ and the corresponding scaling factor $s$ to maximally reduce position bias, which can be formulated as follows:

$$\underset{h_t \in \mathcal{H}, s < 1}{\arg\min} \mathbb{E} \left[ \sum_{i=1}^{|\boldsymbol{P}|} \mathcal{L}\left(\boldsymbol{x}, \boldsymbol{y}, \boldsymbol{p}_i; F(\boldsymbol{\theta}, h_t, s)\right) \right] \tag{2}$$

where $\boldsymbol{P}$ represents the set of different positions of the ground-truth information within the prompt $\boldsymbol{x}$, $F(\boldsymbol{\theta}, h_t, s)$ denotes the operation of scaling the LLM $\boldsymbol{\theta}$ on the $t$-th dimension of its hidden states by the scaling factor $s$, and $\mathcal{L}$ denotes the loss for general downstream tasks of the modified model.

## 3.2 IDENTIFYING POSITIONAL HIDDEN STATES

We have defined positional hidden states in Definition 2.1. However, the original values of hidden states may not strictly satisfy monotonicity. After curve fitting, we can identify dozens or hundreds of dimensions that exhibit various degrees of relevance to positional information. Thus, the first step of our method is to find the dimension that best fits the properties of positional hidden states.

To efficiently search for the positional hidden states from the LLMs' hidden states set, we leverage the characteristics of positional hidden states defined in Section 2.3 and propose a heuristic positional hidden search algorithm. As shown in Algorithm 1, the search process consists of the following two steps: 1) Identify the top-$k$ dimensions $\rho$ in the hidden states that are monotonic in more than $\varepsilon$ layers and are as smooth as possible. Here $c_t$ is the number of layers where $h_t(p)$ is monotonic, and $g_t$ is the smooth score of $h_t(p)$. Equ.(3) is the smoothness formula. 2) Use a small validation dataset (or called calibration dataset) $\mathcal{D}_{\text{val}} = \{\boldsymbol{x}, \boldsymbol{y}\}$ to evaluate the impact of scaling these positional hidden states respectively and select the positional hidden states $h_{\bar{t}}$ that can lead to the minimal loss $\mathcal{L}_{\bar{t}}$.

---

**Algorithm 1** Positional Hidden State Search

1: **Input:** LLM $\boldsymbol{\theta}$, hidden states $\mathcal{H}$, layer number $L$, validation set $\mathcal{D}_{\text{val}}$, positions set $\boldsymbol{P}$, threshold $\varepsilon$

*# Indentify top-K positional dimensions*
2: $\boldsymbol{\rho} \leftarrow \phi$
3: **for** $t \leftarrow 1$ to $|\mathcal{H}|$ **do**
4:     $c_t \leftarrow 0, g_t \leftarrow 0$
5:     **for** $l \leftarrow 1$ to $L$ **do**
6:         **if** $h'_t(p) > 0, \forall\, p$ or $h'_t(p) < 0, \forall\, p$ **then**
7:             $c_t \leftarrow c_t + 1, g_t \leftarrow g_t + \text{Smooth}(h^l_t)$
8:         **end if**
9:     **end for**
10:     **if** $c_t > \varepsilon$ **then**
11:         $\boldsymbol{\rho} \leftarrow \boldsymbol{\rho} \cup \{t\}$
12:     **end if**
13: **end for**
14: $\boldsymbol{\rho} \leftarrow \underset{t \in \boldsymbol{\rho}}{\arg\min_K} g_t$

*# Evaluate on the validation dataset*
15: **for** $t \in \boldsymbol{\rho}$ **do**
16:     $\mathcal{L}_t \leftarrow 0$
17:     **for** $p \in \boldsymbol{P}$ **do**
18:         $\mathcal{L}_t \leftarrow \mathcal{L}_t + \mathcal{L}(\boldsymbol{x}, \boldsymbol{y}, p; F(\boldsymbol{\theta}, h_t, s))$
19:     **end for**
20: **end for**
21: $\bar{t} \leftarrow \underset{t \in \boldsymbol{\rho}}{\arg\min_k} \mathcal{L}_t$
22: **return** $\bar{t}$

---

$$\text{Smooth}(h_t) = \int |h''_t(p)|^2 \tag{3}$$

As for selecting the best scale factor, we take 0.5, 0, -0.5, and -1 to respectively experiment on the validation set, obtain the validation loss, and then select the scaling factor with the lowest loss.

### 3.3 SCALING THE POSITIONAL HIDDEN STATES

To minimize the impact of this modification on the semantics of LLMs, we propose a method scaling the positional hidden states only affecting the last token, as shown in Figure 4. Specifically, for the tokens preceding the last token, the attention calculation remains the same as the original. For the last token's attention computation of a sequence of length $l$, we obtain the modified query state $\overline{q}_l$ (of the $l$-th token, i.e. the last token) and key states $\overline{K}$ (of all the tokens) by scaling the positional hidden states. That is,

$$\overline{q}_l = \mathcal{P}(W^Q f(\boldsymbol{h}(l), p, s), l), \quad \overline{K} = \mathcal{P}(W^K f(\boldsymbol{h}, p, s), [1, 2, ..., l]) \tag{4}$$

Here $f(\boldsymbol{h}, p, s)$ means the $p$-th dimension of $\boldsymbol{h}$ is scaled by the factor $s$. Therefore, the corresponding attention calculation is as follows:

$$\boldsymbol{z} = \begin{cases} \text{Softmax}(\dfrac{\boldsymbol{q}_i \boldsymbol{K}^\top + \text{Mask}}{\sqrt{d}})\boldsymbol{V}, & i < l \\[4mm] \text{Softmax}(\dfrac{\overline{q}_l \overline{K}^\top}{\sqrt{d}})\boldsymbol{V}, & i = l \end{cases} \tag{5}$$

where $\boldsymbol{z}$ is the attention output. We implement our method using FlashAttention (Dao, 2023) with minimal overhead. After calculating the combined attention weights, the remaining computations remain the same as in the original method. As shown in Appendix A.4, our approach results in only a slight increase in latency.

## 4 EXPERIMENTS

### 4.1 SETUP

**Evaluation Tasks and Models** We apply our method to a wide range of state-of-the-art open-source LLMs, including: 1) RoPE (Chen et al., 2023a) models: LLaMA-2 (7B, 13B) (Touvron et al., 2023), Mistral-7B (Jiang et al., 2023a), Gemma-7B (Team et al., 2024), Qwen1.5-7B (Bai et al., 2023a); 2) Context window extended models: Vicuna (7B, 13B) (Chiang et al., 2023); 3) Alibi (Press et al., 2022) models: MPT-30B (Team, 2023). All the models we use are instruction-tuned versions.

And we evaluate the performance across three aspects: 1) Position-bias-related tests on NaturalQuestion multi-document QA (Liu et al., 2024b) and KV retrieval (Liu et al., 2024b) with ground-truth at different positions in the prompt. The NaturalQuestion task includes 20 documents with a prompt length of about 2.3k tokens, while the KV retrieval task includes 140 KV pairs with an average length of about 10k tokens. 2) General long-context benchmark on LongBench (Bai et al., 2023b), including multi-document QA, single-document QA, summarization, few-shot learning, synthetic tasks, and code completion, totaling 16 tasks with an average length of 37k tokens. 3) Position-sensitive tasks on timeline reordering in LooGLE (Li et al., 2023), with an average length of 10k tokens. For prompts that exceed the context windows of LLMs, we follow LongBench's approach by truncating from the middle and retaining the head and tail of the prompt to fit within the context windows. We use the provided metrics and scripts from the following benchmarks for evaluation.

**Implementation Details** In this paper, we implement our approach using PyTorch, HuggingFace Transformers, and FlashAttention (Dao, 2023) in an A100 GPU. To ensure stable and reproducible results, we use greedy decoding in all experiments. For the search part, we set the top-$k$ size of positional hidden states to 10 and $\varepsilon$ to $L/4$, where $L$ is the number of layers. The validation set is a synthetic KV retrieval dataset consisting of 100 examples, which do not overlap with the test set. The search process takes approximately 10 minutes. For the scaling part, we only modify the intermediate layers of the model to minimize the negative impact on performance. The details of the scaling dimensions, layer ranges, and factors are shown in Table 5. More details are provided in Appendix A.

| Methods | NaturalQuestion | | | | | | KV Retrieval | | | | | |
|---|---|---|---|---|---|---|---|---|---|---|---|---|
| | 1st | 5th | 10th | 15th | 20th | Avg. | 0% | 25% | 50% | 75% | 100% | Avg. |
| LLaMA-2-7b-chat | 32.4 | 23.8 | 30.6 | 31.6 | 38.2 | 31.3 | 77.6 | 24.6 | 62.0 | 35.6 | 78.0 | 55.6 |
| LLaMA-2-7b-chat w/ Ms-PoE | **40.8** | 29.2 | 33.0 | 32.8 | 39.6 | 35.1 | **95.0** | 29.8 | 21.4 | **51.8** | 89.8 | 57.6 |
| LLaMA-2-7b-chat w/ Ours | 33.6 | **34.0** | **40.6** | **43.0** | **51.8** | **40.6** | 63.6 | **38.0** | **82.2** | 40.6 | **94.6** | **63.8** |
| LLaMA-2-13b-chat | 45.2 | 39.6 | 40.4 | 44.2 | 51.0 | 44.1 | 74.2 | **39.0** | **70.4** | **84.4** | **86.8** | **71.0** |
| LLaMA-2-13b-chat w/ Ms-PoE | 48.4 | 41.4 | 42.4 | 45.4 | 52.6 | 46.0 | **87.8** | 28.0 | 35.4 | 49.2 | 83.0 | 56.7 |
| LLaMA-2-13b-chat w/ Ours | **50.6** | **43.4** | **45.0** | **49.4** | **58.2** | **49.3** | 41.2 | 17.0 | 49.6 | 76.8 | 84.8 | 53.9 |
| Vicuna-7b-v1.5-16k | **70.4** | 54.8 | 46.8 | 45.8 | 47.8 | 53.1 | **98.4** | 0.8 | 0.2 | 0.2 | 0.2 | 20.0 |
| Vicuna-7b-v1.5-16k w/ Ms-PoE | 67.0 | 55.2 | 50.6 | 46.8 | 48.2 | 53.6 | 97.4 | **36.8** | **15.6** | 5.2 | 6.6 | **32.3** |
| Vicuna-7b-v1.5-16k w/ Ours | 63.8 | **57.6** | **53.6** | **51.2** | **55.6** | **56.4** | 95.4 | 22.0 | 12.6 | **5.2** | **20.4** | 31.1 |
| Vicuna-13b-v1.5-16k | 67.4 | 48.2 | 45.2 | 45.6 | 44.4 | 50.2 | 95.6 | 74.2 | 64.2 | 58.8 | 18.2 | 62.2 |
| Vicuna-13b-v1.5-16k w/ Ms-PoE | **70.0** | 51.4 | 46.8 | 42.8 | 47.0 | 51.6 | 91.8 | 59.4 | 71.6 | **74.4** | **48.8** | 69.2 |
| Vicuna-13b-v1.5-16k w/ Ours | 67.4 | **51.4** | **47.6** | **48.8** | **48.0** | **52.7** | **97.2** | **83.4** | **80.8** | 68.8 | 35.4 | **73.1** |
| Mistral-7b-Instruct-v0.2 | 57.2 | 55.0 | 61.2 | **61.6** | 62.6 | 59.5 | **99.8** | 93.0 | 89.0 | 95.0 | 94.2 | 94.2 |
| Mistral-7b-Instruct-v0.2 w/ Ms-PoE | 58.2 | **60.0** | 62.6 | 58.8 | 62.2 | 60.4 | **99.8** | **95.6** | 88.4 | **96.0** | **95.4** | **95.0** |
| Mistral-7b-Instruct-v0.2 w/ Ours | **61.2** | 56.4 | **63.2** | 59.8 | **64.0** | **60.9** | 97.6 | 93.2 | **90.6** | 95.6 | 93.8 | 94.2 |
| Gemma-1.1-7b-it | 29.6 | 25.2 | 28.2 | 29.6 | 27.4 | 28.0 | **98.6** | 67.0 | 62.4 | 83.4 | **100.0** | 82.3 |
| Gemma-1.1-7b-it w/ Ms-PoE | 33.8 | 29.0 | 31.6 | 28.6 | 28.6 | 30.3 | 0.0 | 0.0 | 0.0 | 0.0 | 0.0 | 0.0 |
| Gemma-1.1-7b-it w/ Ours | **35.4** | **31.4** | **36.0** | **35.4** | **35.0** | **34.6** | 97.6 | **95.8** | **97.6** | **96.8** | 99.6 | **97.5** |
| Qwen1.5-7b-chat | **72.4** | 53.8 | 52.2 | 51.2 | 54.4 | 56.8 | **100.0** | **97.2** | 84.6 | 60.0 | 56.4 | 79.6 |
| Qwen1.5-7b-chat w/ Ms-PoE | 67.4 | 49.8 | 48.2 | 47.4 | 47.0 | 52.0 | 3.4 | 1.4 | 2.8 | 2.6 | 0.6 | 2.2 |
| Qwen1.5-7b-chat w/ Ours | 67.4 | **55.2** | **53.6** | **56.0** | **59.4** | **58.3** | 97.2 | 95.6 | **98.8** | **76.6** | **94.4** | **92.5** |
| MPT-30b-chat | **75.6** | **49.6** | 39.0 | 33.4 | 39.6 | 47.4 | 71.4 | 34.8 | 31.6 | 41.6 | 74.0 | 50.7 |
| MPT-30b-chat w/ Ms-PoE | / | / | / | / | / | / | / | / | / | / | / | / |
| MPT-30b-chat w/ Ours | 75.0 | 48.8 | **41.6** | **40.6** | **44.0** | **50.0** | **99.0** | **65.8** | **48.6** | **46.6** | **69.4** | **65.9** |

Table 1: Performance of different methods with different models on NaturalQuestions (20 docs) (Liu et al., 2024b) and KV retrieval (140 KV pairs) (Liu et al., 2024b) dataset.

**Baselines** We include two training-free positional bias mitigation methods as baselines: (i) **Original**, the unmodified LLM results with the ground truth at different positions in the prompt. (ii) **w/ Ms-PoE** (Zhang et al., 2024), a head-aware position embedding scaling method to mitigate position bias. Following the original settings, we apply scaling coefficients of 1.2 to 1.8 starting from the 3rd layer.

## 4.2 MAIN RESULTS

Tables 1 and 2 present the performance of various methods in different benchmarks. Several observations and conclusions can be drawn: 1) Our method consistently improves overall performance at different positions, with increases of up to 9.3%, 15.2%, and 4.7% in NQ, KV retrieval, and LongBench, respectively, except for LLaMA-2-13B in KV retrieval. Additionally, compared to the SoTA method Ms-PoE, our method shows significant improvements of up to 6.3%, 97.5%, and 14% in NQ, KV retrieval, and LongBench. The poor performance of Ms-PoE in KV retrieval can be attributed to the interpolation causing information loss. 2) Our method effectively enhances LLMs' understanding of information located in the middle and latter parts of the prompt. For key information at the beginning of the prompt, performance is comparable to baselines. Considering only the average performance of the last four positions, our method's improvements over the original increase to 11.3% and 16.8% in NQ and KV retrieval, respectively, and over Ms-PoE increase to 8.7% and 97.5% in NQ and KV retrieval, respectively. 3) Our approach is effective not only for RoPE models but also for context window extended models like Vicuna-16K, which already readjust RoPE (Chen et al., 2023a). Additionally, our method can be adapted to different position embeddings, such as Alibi (Press et al., 2022) models like MPT, resulting in improvements of 2.6%, 15.2%, and 1.2% in NQ, KV retrieval, and LongBench, respectively. 4) Our method demonstrated varying degrees of improvement across different tasks, with the most significant increases being 1.5% in few-shot learning tasks, 3.4% in code tasks, 4% in synthetic tasks, 9.2% in single document QA tasks, and 1.9% in multi-document QA tasks. In summarization tasks, performance was nearly on par with the original results. While our method did not significantly improve the average scores overall, it at least demonstrates that it can mitigate position bias without impairing the model's original capability to handle long context tasks.

| Models | SingleDoc | MultiDoc | Synth. | Summ. | FewShot | Code | AVG |
|---|---|---|---|---|---|---|---|
| LLaMA-2-7b-chat | 28.9 | 29.7 | 6.6 | 26.3 | 61.2 | 47.1 | 33.3 |
| LLaMA-2-7b-chat w/ Ms-PoE | **29.8** | **31.7** | **10.5** | **26.7** | 61.0 | **48.1** | **34.6** |
| LLaMA-2-7b-chat w/ Ours | 29.2 | 29.3 | 9.7 | 25.0 | **61.6** | 46.9 | 33.6 |
| LLaMA-2-13b-chat | 21.4 | 14.6 | 11.2 | 26.1 | 61.5 | 39.8 | 29.1 |
| LLaMA-2-13b-chat w/ Ms-PoE | 20.8 | **15.4** | **12.7** | **27.3** | **62.8** | 36.3 | 29.2 |
| LLaMA-2-13b-chat w/ Ours | **30.6** | 9.6 | 10.8 | 25.7 | 62.6 | **43.2** | **30.4** |
| Vicuna-7b-v1.5-16k | 30.2 | 21.6 | 7.2 | 26.7 | 53.9 | 40.5 | 30.0 |
| Vicuna-7b-v1.5-16k w/ Ms-PoE | **32.3** | **24.2** | 8.3 | **28.0** | **55.2** | **43.1** | **31.8** |
| Vicuna-7b-v1.5-16k w/ Ours | 27.1 | 22.1 | **11.2** | 26.1 | 55.0 | 40.2 | 30.3 |
| Vicuna-13b-v1.5-16k | 31.1 | 33.8 | 21.2 | 26.2 | 62.0 | 39.8 | 35.7 |
| Vicuna-13b-v1.5-16k w/ Ms-PoE | **34.5** | 33.1 | 16.0 | **27.5** | **64.5** | 37.6 | 35.5 |
| Vicuna-13b-v1.5-16k w/ Ours | 30.1 | **35.1** | **25.0** | 25.8 | 63.5 | **41.7** | **36.9** |
| Mistral-7b-Instruct-v0.2 | 37.8 | 28.5 | 49.7 | 28.8 | **65.3** | 52.9 | 43.8 |
| Mistral-7b-Instruct-v0.2 w/ Ms-PoE | **41.7** | 22.2 | 38.4 | 2.8 | 23.8 | 19.5 | 24.7 |
| Mistral-7b-Instruct-v0.2 w/ Ours | 38.4 | **30.4** | **49.8** | **29.4** | 64.8 | 52.9 | **44.3** |
| Gemma-1.1-7b-it | 39.4 | **23.2** | 32.2 | 24.2 | 14.4 | **19.8** | 25.5 |
| Gemma-1.1-7b-it w/ Ms-PoE | **41.7** | 22.2 | **38.4** | **24.9** | 14.0 | 19.5 | **26.8** |
| Gemma-1.1-7b-it w/ Ours | 39.0 | 23.0 | 35.5 | 24.5 | **14.9** | 19.3 | 25.7 |
| Qwen1.5-7b-chat | **46.4** | 39.5 | 38.4 | 22.3 | 56.4 | **50.2** | **42.2** |
| Qwen1.5-7b-chat w/ Ms-PoE | 42.0 | **41.5** | 30.3 | **25.7** | 46.5 | 38.0 | 37.3 |
| Qwen1.5-7b-chat w/ Ours | 45.8 | 38.8 | **38.5** | 22.1 | **57.6** | 49.6 | **42.2** |
| MPT-30b-chat | 27.9 | **21.9** | **7.5** | 25.7 | 57.3 | 39.3 | **29.9** |
| MPT-30b-chat w/ Ms-PoE | / | / | / | / | / | / | / |
| MPT-30b-chat w/ Ours | **29.4** | 19.5 | 6.7 | **25.8** | **57.6** | **40.1** | **29.9** |

Table 2: Performance of different methods with different models on LongBench (Bai et al., 2023b).

## 4.3 ANALYSIS

**From Bias to Balance**  As shown in Table 1, there is an phenomenon that our method mainly benefits when the key information is not at the beginning, but can often decrease performance if the model performs significantly better when the key information is at the beginning. It reveals a possible fact that the positional hidden may be an important factor causing the model to miss the rear parts of the context while focus too much to the beginning parts. Therefore, scaling such dimension can shift the model's attention from being too focused at the beginning to a more balanced distribution. We validated the above points by testing different scale factors, as shown in Figure 5.

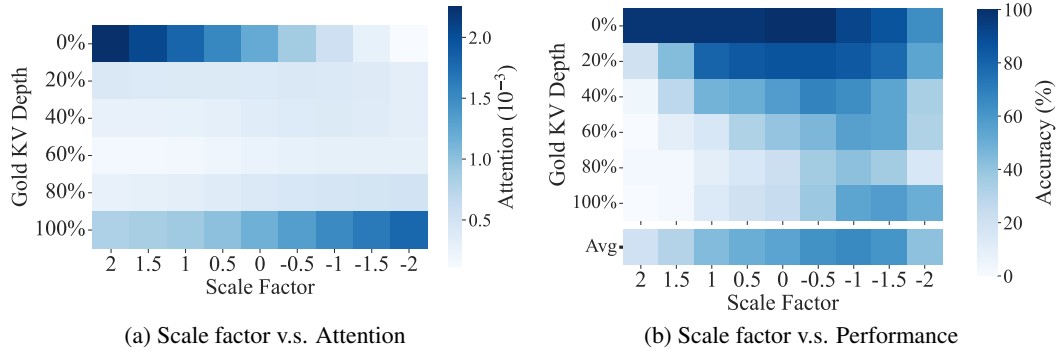

(a) Scale factor v.s. Attention

(b) Scale factor v.s. Performance

Figure 5: Attention distribution and performance when scaling dimension 2393 of Vicuna-7b-v1.5-16k with different scale factors on KV retrieval (Liu et al., 2024b) of 100 KV pairs.

**Scale Factor**  The scaling factor directly controls the degree and direction of the impact of positional hidden states on position bias. As shown in Figure 5, a positive scaling factor causes the model to focus more on the beginning, while a negative factor shifts the focus towards the end. A factor between 0.5 and -1 leads to the most balanced attention distribution, where accuracy also peaks.

These results demonstrate that scaling positional hidden states can influence LLMs' tendency to focus on the beginning, and by adjusting the coefficients, this bias can be effectively mitigated.

| Method | LLaMA-2-7b | Vicuna-13b | Gemma-7b | Mistral-7b | Qwn1.5-7b |
|---|---|---|---|---|---|
| Original | 31.3 | 50.2 | 28.0 | 59.5 | 56.8 |
| Ours | 40.6 | **52.7** | **34.6** | **60.9** | **58.3** |
| w/o monotonicity | 40.6 | 51.8 | **34.6** | **60.9** | **58.3** |
| w/o smoothness | 40.6 | **52.7** | 27.8 | **60.9** | **58.3** |
| w/o validation set | 30.1 | 51.8 | 26.5 | **60.9** | **58.3** |
| w/ scale 2 dimensions | 37.2 | 50.8 | 31.7 | 60.1 | 57.2 |
| w/ modify last 16 tokens | 41.6 | 51.5 | **34.6** | 59.7 | 58.1 |
| w/ modify all tokens | **44.0** | 50.8 | 31.7 | 59.5 | 57.4 |

Table 3: Average performance of different ground-truth positions using different methods on NaturalQuestions multi-document QA dataset (20 docs) Liu et al. (2024b).

**Ablation Study**    To evaluate the contributions of different components in our method, we introduce the following sets for the ablation study: (1) Ours w/o monotonicity, w/o smoothness, and w/o validation set, which adjust the search algorithm by not considering these three indicators, respectively (details in Appendix A.2). (2) Ours w/ scale 2 dimensions, which modifies the top-2 positional hidden states simultaneously. (3) Ours w/ modify last 16 tokens and w/ modify all tokens, which adjust the range of tokens affected by the scaling operation in Equ.(5).

Table 3 shows the ablation results. It can be seen that without filtering by monotonicity or smoothness, performance may decline, and removing the validation set results in more decline in model performance. When the range of tokens or dimensions affected by scaling is expanded, most models experience varying degrees of performance loss. Considering these factors, we choose to modify only the last token and the top-1 positional dimension to achieve the best performance.

**Side Effects**    We utilized the MMLU dataset (Hendrycks et al.), which assesses general capabilities, and the timeline-reorder dataset (Li et al., 2023), which is a task sensitive to positional information, to evaluate whether our approach adversely affects the original abilities of the LLM.

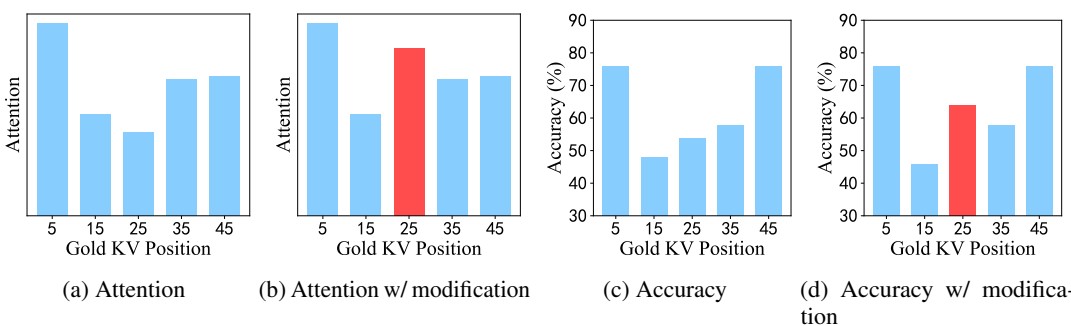

(a) Attention      (b) Attention w/ modification      (c) Accuracy      (d) Accuracy w/ modification

Figure 6: Distribution of attention weight and accuracy as the ground-truth KV is placed at different positions in the prompt. (b) and (d) are situations when the attention on the 25th KV pair is modified.

**Attention v.s. Performance**    As shown in Figure 6, when we manually double the attention weights of the key information (in this case, the 25th KV pair, as illustrated in Figure 6b) during the model's forward pass on the KV retrieval task, the retrieval accuracy for the 25th KV improves, while the accuracy for other parts remains largely unchanged (Figure 6d). This demonstrates that the attention weights for key information are positively correlated with retrieval accuracy.

**Does this Method Compromise the Ability to Perceive Positional Information?**    To demonstrate that our method does not harm the model's performance on general or position-sensitive tasks,

| Model | MMLU | Reorder |
|---|---|---|
| Vicuna-7b-v1.5-16k | 48.22 | 20.83 |
| Vicuna-7b-v1.5-16k w/ Ours | 48.38 | 20.83 |
| Qwen1.5-7b-chat | 60.84 | 28.13 |
| Qwen1.5-7b-chat w/ Ours | 61.43 | 28.13 |
| Mistral-7B-Instruct-v0.2 | 60.31 | 18.75 |
| Mistral-7B-Instruct-v0.2 w/ Ours | 60.38 | 19.79 |

Table 4: Performance of difference models on MMLU and the timeline reorder task.

despite eliminating some positional information, we tested it on two datasets: the MMLU benchmark (Hendrycks et al.) and the timeline reorder task from LooGLE (Li et al., 2023), which involves arranging events chronologically across extensive text. As shown in Table 4, our method does not impair performance on position-sensitive tasks, indicating that the positional information we remove may not be essential for the model's functioning.

## 5 RELATED WORKS

**Long-Context LLMs** Recent research has focused on expanding the context window size of LLMs through four main approaches: 1) Staged pre-training (Nijkamp et al., 2023; Fu et al., 2024), which gradually increases the context window size during training; 2) Modifying or interpolating position embeddings (Press et al., 2022; Chen et al., 2023a; Peng et al., 2023; Ding et al., 2024); 3) Using external memory modules for context storage (Bertsch et al., 2023; Tworkowski et al., 2023); 4) Distributed computation across devices (Liu et al., 2023). While these methods address context expansion, their impact on positional bias in downstream tasks has not been thoroughly explored.

**Addressing Position Bias** Despite explicit positional encoding methods like RoPE (Su et al., 2024) and Alibi (Press et al., 2022), LLMs often exhibit position bias, such as the "lost in the middle" phenomenon (Liu et al., 2024b; Kamradt, 2023). Recent efforts to mitigate this bias fall into several categories: 1) RoPE-based methods: These approaches modify the RoPE computation process to alleviate long-distance information decay, including Attention Bucket (Chen et al., 2023b), which uses an ensemble of multiple RoPE bases to mitigate position bias, and Ms-PoE (Zhang et al., 2024), which dynamically interpolates with a small coefficient for different heads. 2) SFT-based methods (Junqing et al., 2023; Yu, 2023; An et al., 2024): These methods construct data with more diverse key information distributions or employ system2think SFT tasks to mitigate position bias. They require further training of the model. 3) Attention mask-based methods (He et al., 2024): These methods modify attention mechanisms, including Attention Transition (Gao et al., 2023), which redirects attention to significant parts of the context and Stable Mask (Yin et al., 2024), which introduces pseudo attention into the causal mask, ensuring stable attention distribution when facing lengthy texts. 4) Prompt-based methods (Jiang et al., 2023b; Peysakhovich & Lerer, 2023): These methods introduce an external module to reorder or compress information in the prompt, thereby mitigating position bias.

## 6 CONCLUSION

This paper proposes a method for scaling positional hidden states to mitigate position bias issue in LLMs. Specifically, the study first confirms that attention weights manifest position bias within transformers. Additionally, experiments demonstrate that, besides position embeddings, the causal mask also contributes to position bias, which is transmitted to other modules through the hidden states containing absolute positional information, termed as positional hidden states. Based on this, we introduce a prior-based positional hidden search algorithm and mitigate the model's position bias by scaling the positional hidden states searched. Testing eight open-source models with different position embeddings on tasks such as NaturalQuestions Multi-document QA, KV Retrieval, and LongBench, the results show that our method effectively reduces position bias and improves model performance.

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

# A    EXPERIMENT DETAILS

## A.1    DATASETS DETAILS

We choose NaturalQuestion Multi-document QA and Key-Value Retrieval datasets used in "lost in the middle" paper (Liu et al., 2024b) to evaluate the degree to which our method alleviates position bias. NaturalQuestion Multi-document QA require the model to answer the question based on one key information document which is inserted in a long context consisting of many irrelevant documents. And Key-Value Retrieval needs the model to retrieve the value corresponding to the given key from a list consisting of hundreds of Key-Value pairs. These two datasets are both classic in-context tasks which aim to evaluate the differences of model performance when key information is located at different positions in the context. The evaluation metric is accuracy, based on whether the model's response contains a string of the correct answer. In addition, we evaluate our method's improvements across multi task types, using LongBench (Bai et al., 2023b), a benchmark for bilingual, multitask, and comprehensive assessment of long context understanding capabilities of LLMs. It contains six major categories, covering single-document QA, multi-document QA, summarization, few-shot learning, synthetic tasks and code completion. The evaluation metrics are: F1 for single-document QA and multi-document QA, Rouge-L for summarization, accuracy (exact match) for few-shot learning and synthetic tasks, and edit similarity for code completion. During inference, since the original context may sometimes be too long, the input sequences will be truncated in the middle part to avoid exceeding the context window of the model.

## A.2    ADDITIONAL IMPLEMENTION DETAILS

**Curve Fitting**    When we perform curve fitting on $h(p)$, we use least-squares cubic polynomial fit. And when judging its monotonicity, we skip the first 100 positions because the first a few values are often outliers. Since $h(p)$ is originally a discrete function, in practice, we employ the second-order difference to approximate the second-order derivative when computing smoothness.

**Ms-PoE on Mistral**    When applying Ms-PoE (Zhang et al., 2024) to mistral-7b (Jiang et al., 2023a) with its default parameters (minimal scale factor is 1.2 and maximal is 1.8), we found the model fail to generate normal responses, so we set the maximal scale factor to 1.2, under which Ms-PoE (Zhang et al., 2024) is equal to PI (Chen et al., 2023a) with scale factor 1.2.

**Ablation of the Searching Algorithm**    We conducted ablation experiments to demonstrate the necessity of using the three indicators (monotonicity, smoothness, validation loss) in our searching algorithm. Ours w/o monotonicity means we just select top-10 smoothest dimensions and then use the validation loss to determine. Ours w/o smoothness means we select top-10 dimensions with the highest number of monotonic layers and then use validation loss. Ours w/o validation loss means we first select top-10 dimensions with the highest number of monotonic layers and then just choose the smoothest one among them.

## A.3    SCALED DIMENSIONS DETAILS

Table 5: The scaled dimensions, scale factors and applied layers of models.

| Model | Dimension | Scale factor | Applied layers |
|---|---|---|---|
| LLaMA-2-7b-chat | 2,393 | -1 | 10~25 |
| LLaMA-2-13b-chat | 4,283 | -1 | 10~34 |
| Vicuna-7b-v1.5-16k | 2,393 | 0 | 10~25 |
| Vicuna-13b-v1.5-16k | 4,923 | 0 | 10~34 |
| Mistral-7B-Instruct-v0.2 | 213 | 0 | 10~25 |
| Gemma-1.1-7b-it | 1,665 | 0 | 10~22 |
| Qwen1.5-7b-chat | 1,081 | 0.2 | 10~25 |
| MPT-30b-chat | 6,926 | 0 | 10~42 |

The scaled dimensions, scale factors and applied layers of each model we use in out experiments are shown in Table 5.

## A.4 INFERENCE LATENCY

Table 6: Time consumed (minutes) of LLaMA-2-7b-chat in a single A100.

| Method | KV Retrieval | NaturalQuestion |
|---|---|---|
| FlashAttention-2 | 22 | 14 |
| Ours | 32 | 15 |
| Ms-PoE | 61 | 26 |

Table 6 shows the running time of LLaMA-2-7b-chat with different methods in the KV retrieval dataset consisting of 500 samples with average length of about 10,000, and the multi-document QA dataset consisting of 500 samples with average length of about 3,300. Our method requires recompute the query and key states, thus inevitably requires more time compared to baseline, but the cost is within an acceptable range. In contrast, Ms-PoE (Zhang et al., 2024) need to compute the attention weights twice, resulting in a doubling of time consumption.

## B OBTAIN ATTENTION TO KEY INFORMATION

To avoid the influence of internal knowledge in the model and make attention calculation simpler, we conduct a KV retrieval task, whose prompt format is as follows:

> Json data: {"os08jbk1limft6wgxeda": "imx6lyp4b8ogjaq7ret1", ......(n key-value pairs)} The value of key "os08jbk1limft6wgxeda" is "

The last token of the prompt will directly take on the task of predicting the answer, i.e., the value which need to be retrieved. Hence, the last token's attention weights to the previous text can reflect whether it accurately retrieves the key information. We define the model's attention (in some layer) to the key information as $A_G$ in Eq 6, where $G$ represents the set of token positions corresponding to where the key information is at, $l$ is the position of the last token of the prompt, and $a_{l,j}$ represents the attention weight of the $l$-th token to the $j$-th token. By shifting $G$, we use the same method to calculate its attention to each other KV pairs.

$$A_G = \frac{1}{|G|} \sum_{j \in G} a_{l,j} \tag{6}$$

## C HOW WE MODIFY CAUSAL MASK AND POSITION EMBEDDING IN KV RETRIEVAL

In the method 1 in section 2.2, we crop the causal mask to let the "key tokens" unable to attend the previous tokens. As shown in Figure 7, the white part represents the cropped part, which means attention weights are 0, and the orange part represents the attention between tokens within key tokens. In addition, we have retained the attention of key tokens to the first token to maintain the stability of attention distribution. What is more, we only modify the causal mask in layers 1~8, but as the results, the attention to the key information is still significantly improved in layers 15~31, which indicates the positional information generated by causal mask in former layers can be transmitted to latter layers using posisional hidden states as the medium, thus modifying the causal mask solely in the former layers can induce a profound shift in the model's comprehension of positional information.

In the method 2 and 3 in section 2.2, we modify the position embeddings through altering the position ids. The specific operation is shown in the Figure 8, in which we directly replace the position ids corresponding to the key tokens with the position ids of the starting tokens (or the ending tokens) , and actually only the attention weights of the last token to previous tokens are modified. We apply this modification in all the layers. Compared to modifying the causal mask, if only modify position embedding in former layers, the attention in the latter layers remains almost unchanged, which indicates the positional information generated by position embedding may be temporary and can hardly be transmitted across layers.

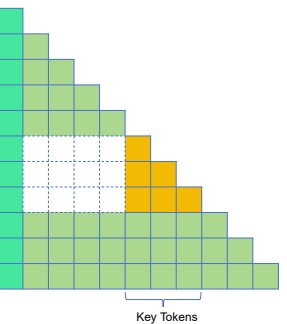

Figure 7: Cropping the causal mask to let key tokens unable to see previous tokens, except the first token.

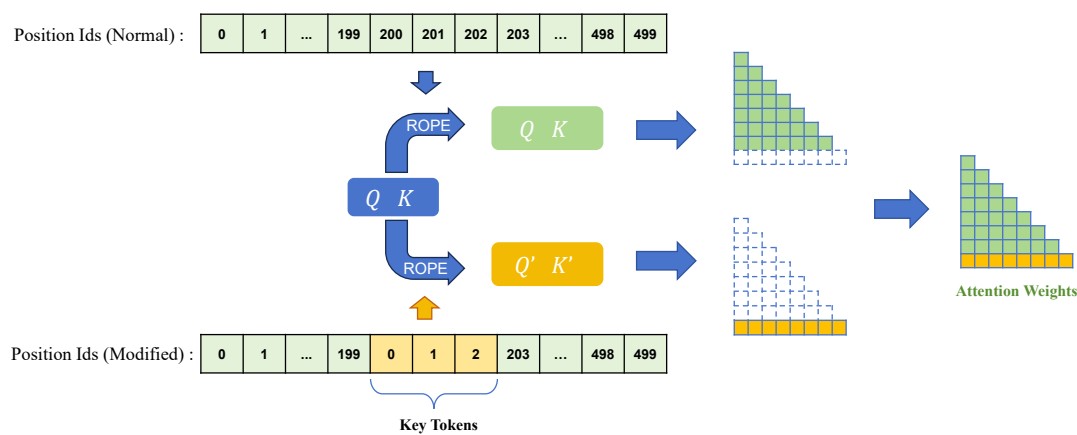

Figure 8: Shifting position ids to the start (PE to beginning).

## D  PERTURBATION ON CAUSAL MASK AND POSITION EMBEDDING

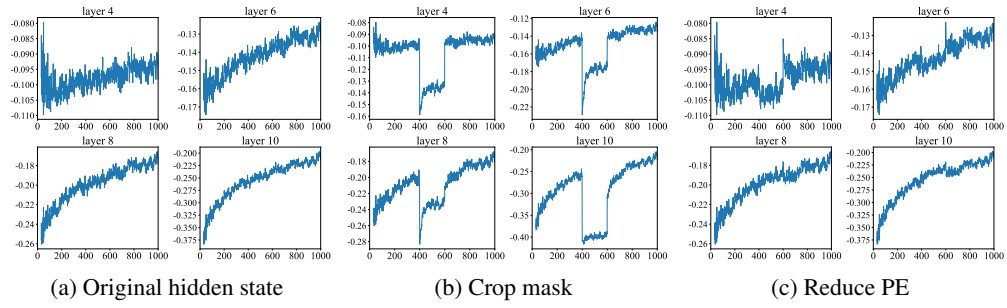

(a) Original hidden state    (b) Crop mask    (c) Reduce PE

Figure 9: We performed perturbation experiments on the causal mask and position embedding (PE), showing the dimension 213 of hidden states of Mistral-7b (Jiang et al., 2023a) using randomly synthesized corpus as input.

To further explore the origin of these position hidden states, we performed perturbation experiments. As depicted in Figure 9c, subtracting 200 from the position ids corresponding to the 400th to 600th tokens (reducing PE) had only a minor effect on the position hidden states, whereas, in Figure 9b, crop the causal mask to make the 400th to 600th tokens unable to attend the 1st to 400th tokens (cropping causal mask) led to significant fluctuations in positional hidden states of the 400th to 600th tokens. This result proves the causal mask is the main factor causing this kind of positional hidden

states, and it is the token's position in the causal mask that determines its value in the positional hidden states, but not position ids of position embedding.

# E   ATTENTION DISTRIBUTION LAYER-WISE AND HEAD-WISE

Figure 10 shows Mistral-7b's attention to each KV pair of each layer (average across all attention heads) in the context in a KV retrieval task when the gold KV is put at different positions. The y-axis is the gold KV's position, x-axis is each KV's position, and the scale of the colorbar represents attention ($10^{-3}$). We can observe that diagonal patterns, which indicates the attention is concentrated on the "key tokens", appear only in the latter layers (start from layer 14), and may be a manifestation of retrieval behavior. In contrast, the former layers only focus on the beginning or end, regardless of where the key information is located.

Figure 11 shows the head-wise situation of layer 15. We can see actually only a portion of attention heads exhibit diagonal patterns, which may correspond to *retrieval heads* (Wu et al., 2024). The attention distribution in these heads also shows a pattern corresponding "loss in the middle", being larger at the beginning or end while significantly smaller at the middle.

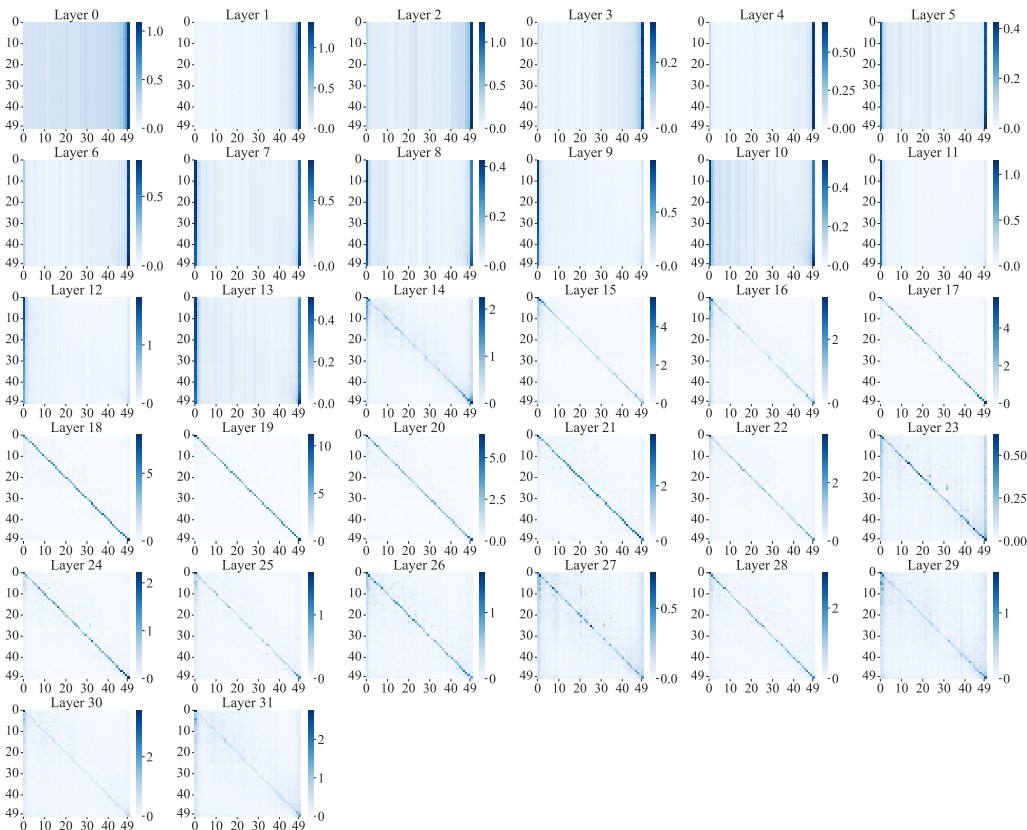

Figure 10: The average attention weight distributed on each KV, of all the 32 layers of Mistral-7b, on a 50 KV pairs retrieval task, when the gold KV is put at each different position.

# F   POSITIONAL HIDDEN STATES VISUALIZATION

We shown various models' positional hidden states of each layer in Figure 12. When visualizing, we discarded the first 30 tokens because the hidden states values of these tokens are often huge (usually hundreds of times larger than the normal value (Sun et al., 2024)), which can disrupt monotonicity. We observed its monotonic trend first appears just in the first layer (actually just after the first attention mechanism), and continue to be more marked.

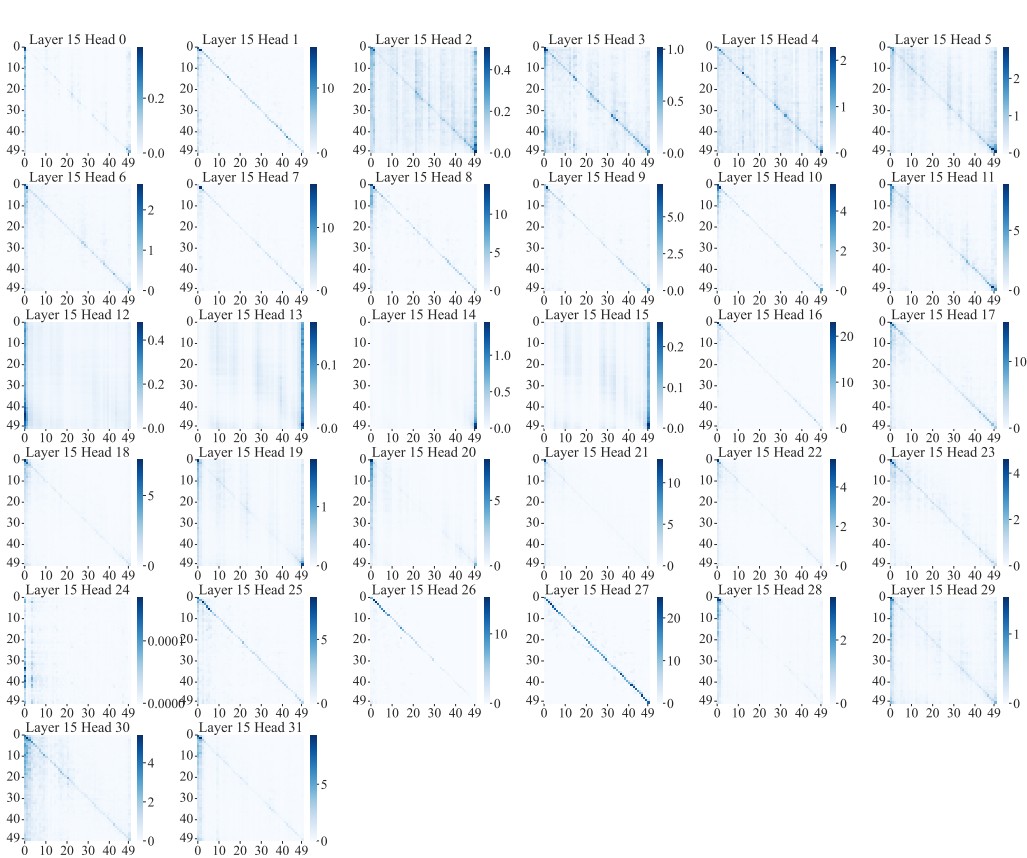

Figure 11: The average attention weight distributed on each KV, of all the 32 attention heads of layer 15 of Mistral-7b, on a 50 KV pairs retrieval task, when the gold KV is put at each different position.

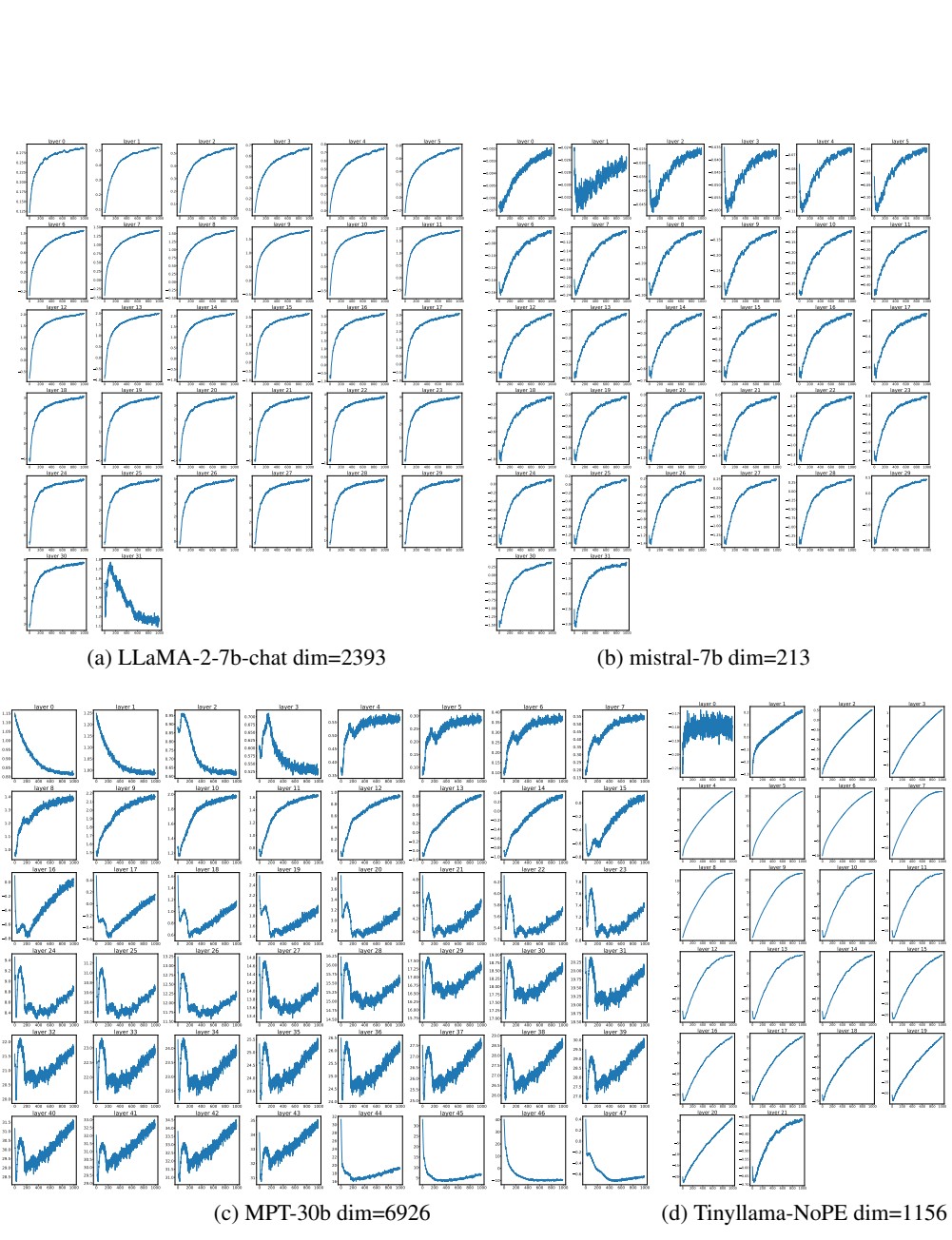

(a) LLaMA-2-7b-chat dim=2393     (b) mistral-7b dim=213

(c) MPT-30b dim=6926     (d) Tinyllama-NoPE dim=1156

Figure 12: Positional hidden states output by each layer of LLaMA-2-7b-chat, Mistral-7b-Instruct-v0.2, MPT-30b-chat and TinyLlama-NoPE-1.1B. The x-axis represents the position, and the y-axis represents the value of the states.

