# OpenReview forum: "Mitigate Position Bias in Large Language Models via Scaling a Single Dimension"
_ICLR.cc/2025/Conference — Submitted to ICLR 2025_

### Official Review · Reviewer_6C5H · 2024-10-27

**Soundness:** 3
**Presentation:** 3
**Contribution:** 2
**Rating:** 6
**Confidence:** 4

**Summary:**

This paper investigates the micro-level mechanisms to address the ``lost of the middle'' phenomenon, which is position bias in LLMs, where information located in the middle of long prompts is underutilized. They figured out that position bias can be reflected in attention patterns and casual mask also introduces position bias. To address this issue, they propose a heuristic search algorithm to find positional hidden states and scales specific hidden state dimensions, denoted as positional hidden states.

**Strengths:**

This paper presents empirical evidence towards positional hidden state.

Proposed method is simple without requiring further training and extensive experiments confirm its effectiveness.

**Weaknesses:**

This paper uses well-known methods for modifying specific hidden states, and there is no methodological innovation although this alone is not a critical issue.

I am not sure why the proposed method is not effective on LongBench. The results show some variations across models and datasets, which would need to be analyzed to assess robustness.

**Questions:**

line 118 in Appendix 4.3 might be Section 4.3?

---

### Official Review · Reviewer_YYdj · 2024-11-01

**Soundness:** 3
**Presentation:** 2
**Contribution:** 2
**Rating:** 5
**Confidence:** 3

**Summary:**

This paper proposes a method for addressing the well-known "lost in the middle" bias for long-context LLMs, in which LLMs struggle to leverage information that is not in the start or end of their prompt. The authors first carry out a thorough investigation which finds that this bias phenomenon can be found in the attention weights produced by the model, not just the accuracy on the downstream performance of a model. Additionally, their analysis provides more evidence that the causal mask allows position embedding less models to identify the position of each token.

Their main method relies on what they call "positional hidden states", which are defined by the authors as dimensions within a model's hidden states that change monotonically across token positions in several of the model's layers. They therefore hypothesize that identifying and optimally adjusting a set of such positional hidden dimensions could reduce the bias against leveraging tokens in the middle.

Their experimental setup finds that this is indeed the case for several open-weight LLMs in both the NQ dataset and KV retrieval. However, the performance differences are much more ambiguous across tasks and models when it comes to other long-context tasks featured in LongBench.

**Strengths:**

- The work is well-motivated and important. It is important for end-users that LLMs have consistent performance regardless of in what order information is presented to an LLM. It is therefore very important to address the lost in the middle problem. In addition, the authors provide a thorough exploration of how attention and causal masks are implicated in creating this order biasing effect, which they then attempt use to motivate their methodology.
-  The experimental setting is quite thorough, with 5 datasets and 8 models involved. The authors also included a recent strong baseline Ms-PoE in their experiments.
- Results on the NQ and KV retrieval are quite promising and the performance on LongBench does not seem to deteriorate significantly.
- The methodology is very creative and technically interesting. The authors hypothesize that some dimension of the model's hidden states is encoding positional bias and this is somewhat confirmed experimentally for simple tasks like NQ and KV retrieval.

**Weaknesses:**

- This paper suffers from lack of clarity. Section 1 and 2 are quite challenging to read, with too many references to the Appendices. Although I believe I now understand their method, it was a challenging process. I highly recommend editing the Figure 4 caption and the figure itself to make the process clearer, i.e. make it explicit that only the last token is scaled and this is why there are multiple colors in the figure.
- The experiment shown in Section 2.2 are a bit problematic, changing the causal mask of only the ground truth KV and point to accuracy increasing due to this as a way to say that the causal mask contributes to the bias is unconvincing.
- I think this paper provides interesting insights into the internal mechanics of how LLMs might be encoding positional information. However, the author's claims that their method can "mitigate position bias by scaling this positional hidden states", which is a bit too strong. Their experiments on LongBench show similar results to the unmodified LLM, indicating that position bias could still be present for tasks other than simple tasks like NQ and KV retrieval. I believe that a deeper dive into LongBench would help the reader understand whether their hypothesis holds generally or only very specifically in simple retrieval tasks.

 - Overall, I believe this paper has value for the community but it needs to address why some of their own experimental evidence goes against their hypothesis. It would also benefit from more polishing.

**Questions:**

- Please let me know if you believe I am misinterpreting the results on LongBench, does the parity with a standard LLM not mean that the bias mitigation was not very successful?

---

### Official Review · Reviewer_mCvw · 2024-11-01

**Soundness:** 2
**Presentation:** 3
**Contribution:** 3
**Rating:** 5
**Confidence:** 4

**Summary:**

The paper addresses the position bias in large language models, particularly the tendency to overlook middle-context information, known as the "lost in the middle" phenomenon. The authors identify that this bias arises from both position embeddings and causal masks, which introduce positional hidden states influencing attention patterns. They propose a mitigation approach by scaling specific dimensions in these positional hidden states, demonstrating its effectiveness across models like LLaMA-2 etc.

**Strengths:**

1. The method shows improvements across various models and tasks, indicating broad applicability.
2. By leveraging FlashAttention and modifying only one dimension, the method remains efficient, with minimal latency impact.
3. The method shows up to 15.2% improvement on position-sensitive benchmarks, suggesting it addresses bias effectively.

**Weaknesses:**

1. Only one hidden state dimension is scaled, which may not capture more nuanced, layer-specific positional dependencies.
2. The impact of position scaling seems to vary by task, suggesting that tuning may be required for optimal results in different contexts.
3. By focusing on single-dimension scaling, the model may become overly specialized to certain bias patterns rather than general long-context processing needs.

**Questions:**

1. How robust is the prior-based search algorithm for identifying positional hidden states across diverse datasets? Could noise in validation data affect the consistency of the identified dimension? How sensitive is the proposed method w.r.t. validation/calibration date?

2. How sensitive is the method to the choice of scaling factor? Would tuning this factor dynamically based on prompt structure (e.g., key information position) enhance retrieval performance?

3. How does the performance of this method compare when combined with techniques like prompt reordering/grounding, rather than as a standalone method?

5. Have you considered modifying multiple layers simultaneously to capture cross-layer dependencies of positional hidden states? How might this impact efficiency and performance?

6. Considering the performance disparities among models such as Mistral and Qwen across various tasks, have you identified any specific characteristics (e.g., number of attention heads, embedding size) that significantly influence the effectiveness of this position scaling method? Based on the outputs, can you classify the LLMs in terms of which models are likely to exhibit higher bias and which are expected to show less?

7. For complex tasks with mixed content (e.g., timeline reordering with nested information), how does the method adapt to multiple levels of positional dependencies within the same prompt? Would modifying additional hidden state dimensions enhance this?

8. Could you discuss the feasibility of extending this approach to handle extremely long contexts beyond the current benchmarks (e.g., 100k tokens or more)? Would the effectiveness of this single-dimension scaling hold, or would adjustments be necessary?

9. How do you think the proposed method would perform with language models that have 1-3 billion parameters (like Phi-2) compared to those with over 10 billion parameters? Additionally, how would quantized versions be affected? Do you anticipate that scaling the proposed algorithm for smaller LMs would yield better performance in terms of bias, even if it results in a slight decrease in accuracy?

---

### Official Review · Reviewer_wmax · 2024-11-04

**Soundness:** 2
**Presentation:** 2
**Contribution:** 2
**Rating:** 3
**Confidence:** 3

**Summary:**

This work analyzes the role of attention mechanism in position biases of LLMs and further proposes a novel method  to mitigate position bias. The main findings include: (1) LLMs' position bias is reflected by attention weights (2) causal masking contributes to position bias and there are certain dimensions in the hidden state that carry information about absolute position. Based on these observations, the authors propose to mitigate position bias by scaling the identified hidden state dimension. Experiments on a variety of tasks and LLMs show that the proposed method might be able to mitigate position bias and improve performance on downstream tasks.

**Strengths:**

1. This paper studies position bias from the angle of hidden states, which I find to be interesting.
2. The experiment setup seems comprehensive, covering a wide range of models and tasks.
3. The analysis and insights could contribute to the understanding of position bias in LLMs.

**Weaknesses:**

1. The finding in Section 2.2 (Causal mask also contributes to position bias) seems trivial to me. The attention mechanism is a core component of Transformer models and naturally plays a significant role in model behaviors. By modifying the attention mask to let the target token only sees itself naturally leads to drastically increased attention weight and KV retrieval performance. The link between Section 2.2 and the main hypothesis on position bias in hidden states seems very weak.
2. The authors introduce a rather strict definition for "positional hidden states", which should demonstrate strict monotonicity. However, the value of identified hidden state dimension (e.g. those in Figure 3) do not fit this strict definition. There also lacks a measure of the degree to which a dimension deviates from strict monotonicity in the paper. Therefore, I can not see how such a definition adds to the contribution of this work.
3. The evaluation result does not show a consistent improvement when applying the proposed method to different LLMs and tasks. Also, since the authors optimize the hyper-parameter for scaling based on the loss on a dev set (Section 3.2), it is also unclear if the performance improvement actually comes from the mitigation of position bias. In fact, I think the scaled attention distribution (Figure 5.(a)) is more skewed toward the 0% and 100% positions than without scaling (Figure 1.(c)).
4. The LLMs tested seem outdated for a submission to ICLR 2025. It is unclear if the proposed method could benefit more recent models with stronger capabilities and longer context window. I strongly suggest updating evaluation results with more recent LLMs, such as Llama-3 series and Gemma-2.

**Questions:**

My major questions have been listed in weaknesses.

One suggestion: In terms of writing, several key setup that are important for understanding the main text are put in the appendix, such as introduction of the KV retrieval task (Appendix B) and the introduction of baselines (Appendix C). I would suggest add more details to the main text for better readability.

---

### Meta-Review · Area_Chair_EXwg · 2024-12-19

**Metareview:**

The paper proposes an approach to identify positional hidden states (defined as dimensions that change monotonically wrt position) and manipulate their scale to address lost in the middle phenomenon. Overall, the reviewers raised several issues, including (1) flawed experiment settings in section 2.2 (2) results are unconvincing as the proposed method does not show consistent improvements across datasets and models.
Moreover, the reviewers raised issues with the clarity of the paper, specifically section 2.

**Additional Comments On Reviewer Discussion:**

NA; authors did not respond to reviews

---

### Decision · Program_Chairs · 2025-01-22

Reject